# Transcriptional Comparison of Genes Associated with Photosynthesis, Photorespiration, and Photo-Assimilate Allocation and Metabolic Profiling of Rice Species

**DOI:** 10.3390/ijms23168901

**Published:** 2022-08-10

**Authors:** Jae-Yeon Joo, Me-Sun Kim, Yong-Gu Cho, Alisdair R. Fernie, Jwakyung Sung

**Affiliations:** 1Department of Crop Science, Chungbuk National University, Cheong-ju 28644, Korea; 2Max-Planck-Institut für Molekulare Pflanzenphysiologie, Am Mühlenberg 1, D-14476 Golm, Germany

**Keywords:** rice species, photosynthesis, photorespiration, sucrose and starch synthesis, sucrose transporter, cell wall synthesis

## Abstract

The ever-increasing human population alongside environmental deterioration has presented a pressing demand for increased food production per unit area. As a consequence, considerable research effort is currently being expended in assessing approaches to enhance crop yields. One such approach is to harness the allelic variation lost in domestication. This is of particular importance since crop wild relatives often exhibit better tolerance to abiotic stresses. Here, we wanted to address the question as to why wild rice species have decreased grain production despite being characterized by enhanced rates of photosynthesis. In order to do so, we selected ten rice species on the basis of the presence of genome information, life span, the prominence of distribution, and habitat type and evaluated the expression of genes in photosynthesis, photorespiration, sucrose and starch synthesis, sucrose transport, and primary and secondary cell walls. We additionally measured the levels of a range of primary metabolites via gas chromatography–mass spectrometry. The results revealed that the wild rice species exhibited not only higher photosynthesis but also superior CO_2_ recovery by photorespiration; showed greater production of photosynthates such as soluble sugars and starch and quick transportation to the sink organs with a possibility of transporting forms such as RFOs, revealing the preferential consumption of soluble sugars to develop both primary and secondary cell walls; and, finally, displayed high glutamine/glutamic acid ratios, indicating that they likely exhibited high N-use efficiency. The findings from the current study thus identify directions for future rice improvement through breeding.

## 1. Introduction

Global food demand is increasing due to concerns such as the rapid increase in world population and climate change. Although crop production continues to increase through breeding and the improvement of cultivation techniques, statistics reveal that it is becoming increasingly difficult to feed the growing population. Moreover, due to global warming, CO_2_ concentrations and annual average temperatures are rising year by year, with climate change-associated crop stress continuing to increase [1,2]. While an increase in CO_2_ concentration can increase the rate of photosynthesis, rising temperatures negatively affect photosynthesis, and may thus cause a lower production of major crops throughout the world [3,4,5]. It is anticipated that environmental changes will lead to a marked reduction (up to 32%) in rice yield and a consequent increase in market prices by up to 78% [6,7]. In particular, the decline of rice production due to rising temperatures is distinct compared to that of other crops [8]. In terms of the preparation for climate change, however, many approaches to improve photosynthesis have been implemented in crop plants, including rice.

The introgression into rice of C_4_ photosynthetic genes, however, recently revealed a small improvement in photosynthesis and even implied that mutants with restricted photorespiration, which is considered a limiting factor to photosynthetic efficiency in rice, also inhibited plant growth due to an accumulation of substrates for enzymes [9,10]. In order to find clues for routes to develop advanced rice varieties, wild rice species, which are a rich source of useful agronomic traits, including increased production and biotic and abiotic stress tolerances, have been extensively explored to understand C_4_ photosynthetic mechanisms, C_3_–C_4_ intermediate anatomical characteristics, water use efficiency and stress tolerance, and, finally, to examine rice yield production [11,12,13,14,15,16,17,18].

Higher biomass production is closely associated with the accumulation of structural carbohydrates, including cellulose, which constitutes 25 to 50% of all plant biomass [19], and this tendency is prominent in wild rice species [20]. Sucrose is well known as a major metabolite transported to the sink from the source, and thus its source–sink fate is determined by the relative strength between vegetative (leaves and roots) and reproductive (grain) organs [21,22]. Therefore, it is likely that the determination of biomass or grain production definitely depends upon the balance of a utilization of sucrose between sink organs. Despite the importance of understanding the metabolic routes cascading from photosynthesis to sucrose allocation in the determination of the rice phenotype, many approaches have, at least partially, been explored for the optimization of biomass and grain yield in rice. Wild rice species display many differences in terms of genetic, physiological, and morphological aspects compared to cultivated rice varieties [23,24,25], including the fact that they display higher photosynthetic performance [26,27] and biotic and abiotic tolerance [28].

Here, we were interested to ask why most wild rice species have relatively lower grain production despite higher photosynthetic capacity [16,18]. To advance our knowledge, this study attempted to compare the differences between rice species, including the cultivated *O. sativa* ssp. *japonica* cv. Saechucheongbyeo (RDA-Suwon433), with a focus on photosynthetic performance, the sucrose-to-starch transition, sucrose transport, and cell wall development via the transcriptional variations in closely involved genes. For this purpose, we selected ten rice species based on basic information such as the presence of genome information, life span, the prominence of distribution, and habitat type (Appendix A). Herein, we report some interesting differences not only with respect to the expression of genes involved in photosynthesis, photorespiration, sucrose and starch synthesis, sucrose transport, and primary and secondary cell wall synthesis but also in targeted metabolites.

## 2. Results

### 2.1. Growth and Photosynthetic Performance in Rice Species

We implemented the study with ten rice species, *O. sativa* ssp. *japonica* cv. Saechungbyeo (AA, RDA-Suwon433, Korea), *O. glaberrima* (AA, IRGC 96864), *O. nivara* (AA, IRGC 105920), *O. meridionalis* (AA, IRGC 105302), *O. rufipogon* (AA, IRGC 106274), *O. punctata* (BB, IRGC 105690), *O. minuta* (BBCC, IRGC 105124), *O. officinalis* (CC, IRGC 105105), *O. alta* (CCDD, IRGC 105143), and *O. latifolia* (CCDD, IRGC 99585), which have remarkable differences in overall growth and architectural features at both vegetative and reductive stages (Figure 1). All of the wild species except *O. officinalis* and *O. latifolia* have longer-awned seeds compared to the cultivars, and the length of awn was visually noticeable in the AA genotype (Figure 1). The heading date from transplanting ranged from 60 to 133 days, with the shortest period in *O. sativa*, being half of that compared to most of the other rice species (Figure 1). By contrast, days from heading to ripening (seed harvest) were variable (28~39 days). To tease apart the variations resulting from distribution and genotype, we will, hereafter, focus on comparing wild species showing distinct features, and the detail on the basis of their prominent distribution (continent) and genotype is in the Appendix A. Plant height and leaf length were invariant across continent and genotype, whereas tillering was significantly much higher in Asian species and leaf width was markedly greater in American species and the CCDD genotype which displayed greater than two-fold wider leaves compared to the other groups (Appendix A). The highest height tillering, leaf length, and leaf width were observed in *O. rufipogon* and *O. alta*, respectively (Figure 2).

### 2.2. Photosynthesis and Photorespiration in Rice Species

Photosynthetic performance and water use efficiency (WUE, *P_n_/E*) were measured in the upper fully expanded leaves at 60 DAT (Figure 3). In general, wild rice species displayed a superior photosynthetic capacity to other genotypes. In particular, rice species *O. punctata* (BB) and *O. latifolia* (CCDD) revealed remarkable photosynthetic performance, exhibiting 1.3~1.6-fold greater rates (11.35 ± 0.34 and 12.11 ± 0.34 μmol m^−2^ s^−1^) compared to *O. sativa* (8.49 ± 1.23) and *O. glaberrima* (7.49 ± 1.92). In terms of WUE, *O. minuta* (BBCC), showing higher intercellular CO_2_ concentration and lower transpiration rate (*E*), was also considered an interesting wild rice resource. The net photosynthetic rates (*P_n_*) did not represent significant differences by continent, whereas, by genotype, BB and CCDD were markedly higher compared to other groups (Appendix A). Stomatal conductance (*g_s_*) and intercellular CO_2_ concentration were not different between continents and genotypes. Transpiration rates (*E*) showed a similar pattern to CO_2_ fixation rates, which were higher in BB and CCDD and in American species. *O. minuta* (BBCC genotype) was the highest WUE (*P_n_/E*), indicating more than three.

To evaluate the differences in photosynthesis and photorespiration characteristics of rice species, eight main enzymes (four for photosynthesis and four for photorespiration) were selected, and the relative expression patterns of genes encoding those were investigated using qRT-PCR (Figure 4 and Appendix A). An African species has a higher expression of the oxygen-evolving complex (OEC, *Os07g0544800*), which is the very first enzyme of the electro-transfer reaction, compared to other groups. Namely, the expression level of gene-encoding OEC in *O. punctata* (BB genotype) was revealed to be 3~5 times greater than other groups. The expression levels of the representative genes of three enzymes, namely, rubisco-1,5-bisphosphate carboxylase/oxygenase (Rubisco, *Os12g0274700*), phosphoglycerate kinase (PGK, *Os05g0496200*), and phosphoribulo kinase (PRK, *Os02g0698000*) employed in the Calvin–Benson cycle did not differ between continents, but the CC genotype (*O. officinalis*) displayed significantly higher expression levels of all three genes compared to other genotypes. Interestingly, the expression of genes encoding Rubisco (3.96~12.42-fold higher), PGK (1.54~25.94-fold higher), and PRK (1.02~6.95-fold higher) were remarkably higher in all wild species compared to *O. sativa* (Asian cultivated species) (Figure 4). Photorespiration is often considered a mechanism limiting photosynthesis [29], and thus high rates have been anticipated to restrict the growth and yield of C_3_ crop plants. By contrast, glycine decarboxylase (GDC, *Os06g0611900*), which is a key enzyme in the photorespiration cycle, directly affected net photosynthetic rates [10,30]. That said, a few approaches have recently shown that increasing the activities of photorespiratory enzymes actually enhances the rate of photosynthesis. Irrespective of the direction of change, the rate of photosynthesis is altered, and hence, evaluating the expression levels of the photorespiration-related enzymes, phosphoglycolate phosphatase (PGLP, *Os04g0490800*), glycolate oxidase (GOX, *Os07g0616500*), glutamate glyoxylate aminotransferase (GGAT, *Os07g0108350*), and glycine decarboxylase (GDC), is also of interest. The habitat did not affect expression levels of genes associated with photorespiration, and only GGAT showed a significant increase in the BB genotype (*O. punctata*).

### 2.3. The Sucrose-to-Starch Transition and Sucrose Transport in Rice Species

As a result of photosynthesis, sucrose and starch are synthesized and then transferred from source to sink or temporarily stored as starch in the source. Overall, in most wild rice species, it seemed that sucrose synthesis was the relatively predominant route compared to starch accumulation in terms of the relative expression levels of genes with sucrose synthase (SS, *Os06g0194900*). To confirm this, the concentrations of soluble sugar and starch were measured from the four selected rice species (Figure 5). Soluble sugar was higher in all selected species, whereas starch depended on species; significantly higher in *O. nivara* and *O. latifolia* and lower in *O. punctata*. Interestingly, several wild rice species represented higher transcripts of proteins tasked with transporting ADP-glucose into chloroplasts, along with the significantly lower expression of a representative gene encoding starch branching enzyme (BEIIa, *Os04g0409200*), which is responsible for amylopectin biosynthesis (Figure 5). The expression of seven genes encoding enzymes closely involved in the synthesis and utilization of photo-assimilates were invariant across continents, with the exception of genotypes, except for genes encoding sucrose phosphatase (SPP, *Os01g0376700*) and starch branching enzyme (BEIIa) (Appendix A). In addition, on the basis of transcript levels, it appears that *O. rufipogon* (AA) was extremely well primed for sucrose synthesis and transportation (Figure 5), even though most wild species also displayed a noticeable potential in this direction.

### 2.4. Primary and Secondary Cell Wall Synthesis in Rice Species

One possibility of higher sucrose outflow from the leaf mesophyll is its use as a source for cell wall synthesis, and thus the relative expression levels of five and three genes encoding enzymes closely associated with primary and secondary cell wall development, respectively, were compared (Figure 6). Interestingly, wild rice species revealed a substantially higher expression of most of the genes involved in both primary and secondary cell wall development. Of the three main routes of primary cell wall synthesis, AA (*O. nivara*, *O. meridionalis*, and *O. rufipogon*) and CC (*O. officinalis*) genotypes preferentially utilized the Csl C (*Os09g0428000*)—Endoglucanase (*Os06g0247900*)—XTH (*Os10g0577500*) cascade. *O. punctate* (BB) and *O. minuta* (BBCC) seemed to employ both the Csl C—Endoglucanase—XTH and the GT43 (*Os05g0123100*) pathways. Finally, the CCDD (*O. alta* and *O. latifolia*) genotype equally modulated the two pathways described above and that featuring the GT8 (*Os03g0678800*) enzyme.

### 2.5. Correlation between Photosynthesis, Growth Parameters, and Gene Expression

The correlation analysis was performed on the basis of the observations, photosynthetic parameters, growth parameters, and gene expressions in relation to photosynthesis/photorespiration, sucrose/starch synthesis, and cell wall development (Figure 7), and the positive relationship between parameters was observed. Of those, leaf width, SS, and ADP2a (*Os08g0345800*) were strongly positive with cell wall development genes such as Endoglucanase, XTH, GT43, GT8, GPAT3, and BAHD. Photosynthetic genes, PGK and PPK, were greatly positive with SPP. The sucrose synthetic gene, SS, was also positive with SUT1 (*Os03g0170900*) and AGPase (*Os08g0345800*). By contrast, the starch synthetic gene, BEIIa, showed a negative relationship with several cell wall development genes such as XTH, FAR (*Os09g0567500*), and ABC (*Os03g0281900*).

### 2.6. Metabolite Profiling of Rice Species

The fifty targeted metabolites using GC-TOFMS were measured from the upper fully expanded leaves of 10 rice species (Figure 8). Soluble sugars, including sugar-alcohols, differed between rice species, and, of those, mannose and galactose represented relatively higher levels compared to *O. sativa*. Wild rice species revealed an abundance of amino acids such as tryptophan, cysteine, lysine, β-alanine, glutamine, and putrescine, and secondary metabolism intermediates, ferulic and sinapinic acids. Meanwhile, essential amino acids including glycine and serine, directly involved in the photorespiration process, showed a relatively lower accumulation. Additionally, the glutamine/glutamic acid ratio, an indicator of nitrogen use efficiency (NUE), was totally higher in wild rice species, especially *O. merdionalis* (5.53 ± 1.87) and *O. latifolia* (3.67 ± 0.52) (*p* < 0.05).

## 3. Discussion

Rice species are geographically distributed from tropical to temperate areas with different altitudes, light dependence, and water requirements, i.e., *O. nivara* grows under wet and dry seasons [23,24] and *O. rufipogon* grows in permanent wet conditions [24]. We investigated the differences in the growth, photosynthetic activity, production and utilization of photosynthates, and cell wall development between ten rice species, two cultivated and eight wild germplasms. The tillering is species-specific as a botanical characteristic [31], and thus is considered a consequence of genetic variation during acclimating to habitat environments. The difference in tillers was not observed in our study, however, wild rice species which are generally perennial, show an extended tillering duration compared to the cultivated species (Figure 1 and [32]), and this is likely to result in different uses of photosynthates. Furthermore, leaf width is broader in rice species that have genotypes, CC and CCDD, mostly distributed in America (Appendix A). Zhao et al. [18] also reported that CC and CCDD genotypes had greater leaf area compared to other groups, AA, BB, EE, and BBCC.

The CO_2_ fixation rate (*P_n_*) was the highest in *O. latifolia*, and *O. punctata* and *O. minuta* were significantly higher compared to the African cultivated species, *O. glaberrima*. However, this is clearly supported by previous works that reported that photosynthesis in wild rice species was superior to *O. sativa*, although there were a few variations between wild rice species [18,28,33,34,35,36]. Tsutsumi et al. [35] explained that a higher photosynthetic rate in *O. latifolia* was due to bigger mesophyll cells and greater Rubisco activity, and the photosynthetic rate showed a positive correlation with leaf width [35,37]. Water use efficiency (WUE, photosynthetic rate, *P_n_*/transpiration rate, *E*) is often defined as an indicator of drought resistance, and some wild rice species, *O. meridionalis*, *O. officinalis*, and *O. punctata*, were known as drought-resistant species [38]. In this study, *O. minuta*, shade- to semi-shade-grown, exhibited significantly higher WUE, and our finding suggests that *O. minuta* could be a useful source of germplasm to enhance WUE against climate change [39].

Indeed, wild rice species seemed to more efficiently operate the Calvin cycle in terms of transcript levels encoding Rubisco, PGK, and PRK compared to the cultivated rice, *O. sativa*. Interestingly, *O. officinalis*, CC genotype and perennial, was distinguished from other rice species (Figure 4). Information on Calvin cycle-involved enzymes, including Rubisco in wild rice species, is quite limited, and thus, our findings are valuable for further work. Photorespiration in C_3_ plants represents the duality of limitations in photosynthesis [40] and protection from photooxidative damage [41]. Contrary to expectations, some wild species, including *O. rufipogon*, showed higher transcript levels of photorespiration-related enzymes. Phosphoglycolate, a strong inhibitor of photosynthesis, generated by the incorporation of O_2_ into RuBP, was rapidly transformed into glycolate by PGLP [10,42]. However, the lower expression of gene-encoding PGLP resulted in a significant reduction (approximate 40%) in photosynthesis [43]. Therefore, it is suggested that *O. rufipogon* is effective to minimize the inhibition of photosynthesis due to photorespiration. Additionally, GDC releases CO_2_ during the decarboxylation of glycine and serine. The 75% of CO_2_ regenerated during photorespiration enters toward the Calvin cycle [42,44], and this circuit finally plays a key role in lowering the CO_2_ compensation point [45,46,47]. Higher photorespiration results in a greater glycine/serine ratio [48,49]; however, the current study did not exhibit a significant difference in this ratio between rice species (Figure 8a). Photosynthesis strongly depends on GDC activity; reduction in *gdc* mutant in *O. sativa* [30] and enhancement in *GDC* overexpression in *A. thaliana* [50]. Thus, at least, *O. nivara*, *O. rufipogon*, and *O. punctata* are likely to provide clues to improve the interdependence between photosynthesis and photorespiration within rice species. The result of photosynthesis is the synthesis of soluble sugars and starch, and the relatively higher photosynthetic rates in wild species showed a trend in the greater accumulation of soluble sugars (not significant) and starch (significant abundance in *O. nivara*, *O. meridionalis*, and *O. latifolia*) (Figure 5). Interestingly, sucrose synthesis by SS is preferential in most wild species except for *O. officinalis* (SPS), and sucrose was rapidly transported by OsSUT1 rather than stored in the vacuole (Figure 5). Zhang et al. [51] reported that higher transcript levels in *OsSUT1* and *OsSUT2* led to an efflux of soluble sugars from the leaves to the sinks. The transcript level of *OsSUT1* in the flag leaf of *O. australiensis* was significantly higher compared to *O. sativa*, and, finally, resulted in a huge accumulation of sucrose in phloem sap [20]. Similar trends in gene expression were observed in our study, and, more interestingly, came from individual soluble sugars. Essential sugars such as glucose, fructose, and sucrose did not differ between rice species. On the other hand, mannose and galactose in most wild rice species were obviously abundant. Both sugars were higher in *O. longistaminata* compared to *O. sativa* [52]. These sugars might be a predominant form which is common in wild rice species, and, if so, it raises the very important possibility that wild rice facilitates not only sucrose but also raffinose family (RFOs) transportation, supporting that raffinose in *O. longistaminata* was significantly less in the leaf but accumulated in the stem [52]. Besides the rapid transportation of soluble sugars toward the sink, accelerated glycolysis is likely to be a common phenomenon to be converted to other organic compounds including amino acids in wild rice species (Figure 8a,b). Starch synthesis in wild rice species is also likely to be predominant (Figure 5), and it was reported that several wild rice species highly accumulated starch [20,53]. In this study, gene expression encoding AGPase, converting G1P to ADP-Glc, was significantly higher in *O. meridionalis* and *O. latifolia*, whereas genes directly involved in SSIIb and BEIIa did not differ or were significantly lower. Based on obtained results, our suggestion is that the deposition of chloroplast starch might be terminated early due to the predominant accumulation, and G1P could be turned around toward cytosol.

The fact that soluble sugars in wild rice species are quickly lost brings out the possibility that they are consumed for cell elongation, cell wall development, the emergence of new leaf and tiller, etc. It was confirmed that wild rice species exhibited significantly higher photosynthesis and biomass, whereas they showed lower seed production compared to the cultivated rice [20], and they showed a tendency to consume energy, soluble sugars, to cope with biotic and abiotic stresses rather than for seed production [54]. Most genes encoding enzymes associated with primary and secondary cell developments were higher in *O. meridionalis*, and *O. alta*. These wild species were likely to differently facilitate both cell wall developments. For primary, the cellulose-preferential pathway in *O. meridionalis* and glycosyl-derived in *O. alta*; for secondary, both glycerol and fatty acyl dominated in *O. meridionalis* and fatty acyl in *O. alta*, respectively. Indeed, both species revealed higher levels of lignin precursors, ferulic, and sinapinic acids (Figure 8a), and all wild rice species showed a great abundance in tryptophan, which is a precursor of cell growth-promoting hormone, auxin. The authors of [52] also reported that *O. longistaminata* showed higher IAA content compared to the cultivated species, and thus, it is assumed that the higher level of tryptophan in wild rice species is closely associated with greater biomass production, including leaf morphology and tillering. We also have taken valuable findings from the targeted metabolite profiling of wild rice species (Figure 8a,b). Of those, wild rice species have a higher glutamine/glutamic acid ratio, which indicates high nitrogen use efficiency (NUE). As a measure of improving NUE, some works tried to perform the overexpression of the gene involved in glutamine synthase (GS) [55] and select higher GS activity lines under lower N conditions [56]. Secondly, wild rice species accumulate higher levels of putrescine, functioning as temporary nitrogen storage and a reactive oxygen species (ROS) scavenger. With the requirement of further study, it might be suggested that high putrescine in wild rice species is to efficiently use limited N and/or to adjust to adverse environments.

In conclusion, we have obtained valuable information from wild rice species as a measure to improve the efficient use of natural resources, enhance crop yield for feeding the increasing global population, and strengthen the resistance/tolerance against biotic and abiotic stresses. Wild rice species (i) exhibited not only higher photosynthesis but also superior CO_2_ recovery by photorespiration, (ii) showed the greater production of photosynthates such as soluble sugars and starch and quick transportation to the sink organs with a possibility of transporting forms such as RFOs, (iii) revealed the preferential consumption of soluble sugars to develop both primary and secondary cell walls, and (iv) displayed a high glutamine/glutamic acid ratio, indicating high N-use efficiency. The findings from the current study thus identify directions for future rice improvement through breeding.

## 4. Materials and Methods

### 4.1. Plant Material and Sample Preparation

Ten rice species including cultivated rice (*O. sativa* cv. Saechucheongbyeo, RDA-Suwon433) were used for the study. Nine wild rice species (IRGC 96864, 105920, 105302, 106274, 105690, 105124, 105105, 105143, and 99585) were introduced to Chungbuk National University (CBNU, Cheongju, Korea) by a Seconded Special Material Transfer Agreement (seconded SMTA), via Hankyong National University (HKNU, Anseong, Korea), with the IRRI (Los Baňos, Philippines). The seeds of all wild rice species were formally obtained from IRRI. Firstly, to break dormancy, seeds were dried for 48 h at 50 °C, primed with 0.75% KNO_3_ for 48 h at 25 °C, and further dried for 48 h. After surface sterilization (2% of sodium hypochlorite) to easily remove the seed coat, they were placed in an incubator at 25 °C in darkness until germination. As coleoptiles emerged, the seedlings were transferred into containers (10 × 15 × 10 cm, seedling/container) and were subsequently grown in a controlled growth chamber adjusted to 25/20 °C (day/night), 75% RH, and 14 h/10 h (day/night). Rice seedlings (3rd leaf stage) were carefully transplanted into containers (20 × 30 × 20 cm, seedling/container) and grown in a greenhouse under natural climate conditions. In order to avoid the shortage of water and nutrients, water and nutrients (1/3-strength Hoagland solution) were supplied every day and every week, respectively. To compare transcriptional variations between rice species, the upper fully expanded leaves from 3 to 4 independent rice plants were carefully taken, and were immediately frozen in liquid N_2_ for further analysis.

### 4.2. Phenotypic Characteristics and Photosynthesis

Due to the different growth trends of the various rice species, vegetative agronomic traits such as plant height, tillering, and leaf length and width were measured at 60 days after transplanting (DAT). The photosynthetic rate (P_N_), stomatal conductance (g_s_), intercellular CO_2_ concentration (ci), and transpiration rate (E) were measured between 10:00 and 12:00 with the fully expanded leaves at the maximum tillering to panicle formation stage (60 DAT, based on *O. sativa* cv. Saechucheongbyeo) using a portable photosynthesis analyzer (ADC, LCpro+, ADC BioScientific Ltd., UK), which is equipped with a square chamber (6.25 cm^2^). The photosynthetic quantum flux density (PPFD) from solar radiation at the time of assay was 1000 ± 100 μmol m^−2^ s^−1^, and the CO_2_ level was 400 ± 10 μmol m^−2^.

### 4.3. RNA Isolation, cDNA Synthesis, and Quantitative Real-Time PCR Analysis

Total RNA from the leaves of 10 rice species was extracted using TRlzol reagent (Invitrogen, Carlsbad, CA, USA) according to the manufacturer’s protocol. The purity and concentration of the extracted RNA were estimated using NanoDrop (Thermo Fisher Scientific, Madison, WI, USA) and checked on 1.2% agarose gel. The total RNA from the leaf blades was extracted using a Total RNA Extraction Kit (Intron Biotech., Seongnam, Korea) according to the manufacturer’s instructions. A first-strand synthesis performed using a Maxime RT PreMix Kit with Oligo (dT) primer was used for cDNA synthesis from 1 μg of total RNA. The qRT-PCR was performed to analyze the relative gene expression of RNA using EvaGreen Q Master (LaboPass, Seoul, Korea) with a CFX Connect Optics Real-Time System (Bio-Rad, Hercules, CA, USA) according to the manufacturer’s instructions. A quantification method (2^−ΔΔCt^) was used and the variation in expression was estimated using three biological replicates. The rice actin gene was used as an internal control to normalize the data. The PCR conditions consisted of an initial denaturation step at 95 °C for 25 s, followed by 60 cycles of denaturation at 95 °C for 10 s, and annealing and extension at Tm [AW1] (°C) designated by each of the primer sets for 5 s.

### 4.4. Extraction and Analysis of Polar Metabolites

Polar metabolites were extracted as described previously [57]. The metabolites were extracted from powdered tissue (100 mg) by adding 1 mL of 2.5:1:1 (v/v/v) methanol:water:chloroform. Ribitol (60 µL, 0.2 mg/mL) was used as an internal standard (IS). Extraction was performed at 37 °C at a mixing frequency of 1200 rpm for 30 min using a Thermomixer Compact (Eppendorf AG, Hamburg, Germany). The extracts were centrifuged at 16,000× *g* for 3 min. The polar phase (0.8 mL) mixed with 0.4 mL water was centrifuged at 16,000× *g* for 3 min. The methanol/water phase was dried in a centrifugal concentrator (CC-105, TOMY, Tokyo, Japan) for 2 h, followed by a freeze dryer for 16 h. MO derivatization was performed by adding 80 μL of methoxyamine hydrochloride (20 mg/mL) in pyridine and shaking at 30 °C for 90 min. TMS-esterification was performed by adding 80 μL of MSTFA, followed by incubation at 37 °C for 30 min. GC-TOFMS was performed using an Agilent 7890A gas chromatograph (Agilent, Atlanta, GA, USA) coupled to a Pegasus HT-TOF mass spectrometer (LECO, St. Joseph, MI, USA). Each derivatized sample (1 µL) was separated on a 30 cm × 0.25 mm I.D. fused-silica capillary column coated with 0.25-µm CP-SIL 8 CB low bleed (Varian Inc., Palo Alto, CA, USA). The split ratio was set to 1:25. The injector temperature was 230 °C, and the flow rate of helium gas through the column was fixed at 1.0 mL/min. The temperature was set up as follows: initial of 80 °C for 2 min, followed by an increase to 320 °C at 15 °C/min and a 10 min hold at 320 °C. The transfer line temperature and ion-source temperature were 250 and 200 °C, respectively. The scanned mass range was 85–600 *m/z*, and the detector voltage was set to 1700 V. ChromaTOF software was used to support peak findings prior to quantitative analysis and for the automated deconvolution of the reference mass spectra. NIST and in-house libraries for standard chemicals were utilized for compound identification. The calculations used to quantify the concentrations of all analytes were based on the peak area ratios for each compound relative to the peak area of the IS.

### 4.5. Concentrations of Total Soluble Sugars and Starch

In order to determine carbohydrate contents, the dried leaf samples (0.2 g) were first boiled with 10 mL of 80% EtOH in boiling water. The alcoholic extracts were evaporated under a nitrogen stream, and the residues were re-dissolved with distilled water. The residue was digested with 2 mL of 9.3 N HClO_4_, and the supernatant after centrifugation was used for the determination of starch. Water extracts were mixed with 2 volumes of 0.2% anthrone in a concentrated H_2_SO_4_ followed by the estimation of carbohydrates as described by [58]. Glucose was used as a standard for both soluble sugars and starch (Appendix A).

### 4.6. Statistics

The data were analyzed using analysis of variance (ANOVA) followed by Tukey’s HSD test. Data analyses were performed using RStudio (v.4.0.3.). Heatmaps were constructed with RStudio using packages “g-plots” and “p-heatmap”. The correlations between data were evaluated using Pearson’s correlation analysis.

## Figures and Tables

**Figure 1 ijms-23-08901-f001:**
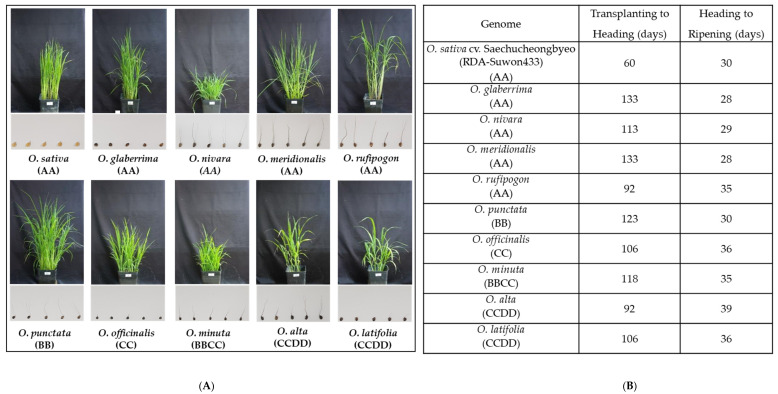
Phenotypic differences in shoot growth and seed between rice species (**A**) and the duration to heading or ripening (**B**). Rice seedlings were transplanted at the 3rd leaf stage and the images of rice shoots were obtained at the maximum tillering to panicle formation stage (60 days after transplanting, based on *O. sativa* cv. Saechucheongbyeo). The harvest of seeds differed from species due to different flowering times.

**Figure 2 ijms-23-08901-f002:**
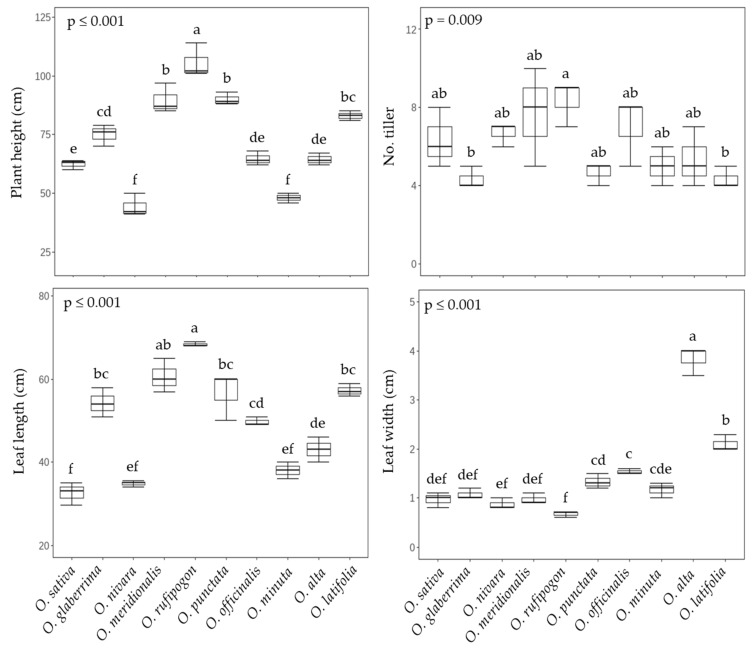
Comparison of vegetative agronomic traits between rice species at 60 DAT (mean of three replications ± S.D.). Means followed by different letters in the same raw are significantly different at *p* < 0.05 according to Turkey’s HSD test.

**Figure 3 ijms-23-08901-f003:**
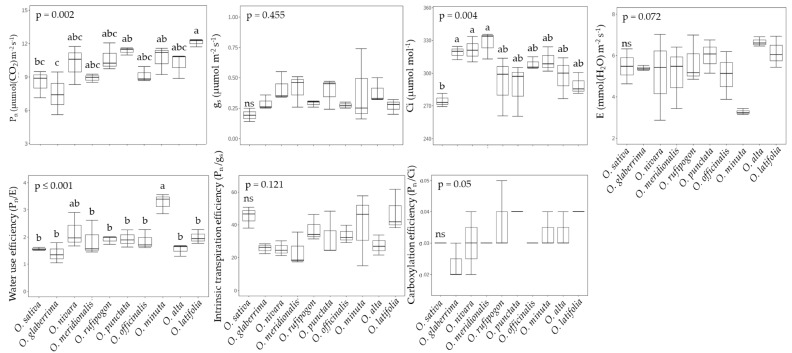
Variations in leaf photosynthetic characteristics water use efficiency (WUE) between wild rice species at 60 DAT (mean of three replications ± S.D.). *P_n_*—net photosynthetic rate (μmol (CO_2_) m^−2^ s^−1^); g_s_—stomatal conductance (μmol (H_2_O) m^−2^ s^−1^); *Ci*—intercellular CO_2_ concentration (μmol mol^−1^); *E*—transpiration rate (mmol (H_2_O) m^−2^ s^−1^); WUE (*P_N_/E*)—water use efficiency. Means followed by different letters in the same raw are significantly different at *p* < 0.05 according to Tukey’s HSD test.

**Figure 4 ijms-23-08901-f004:**
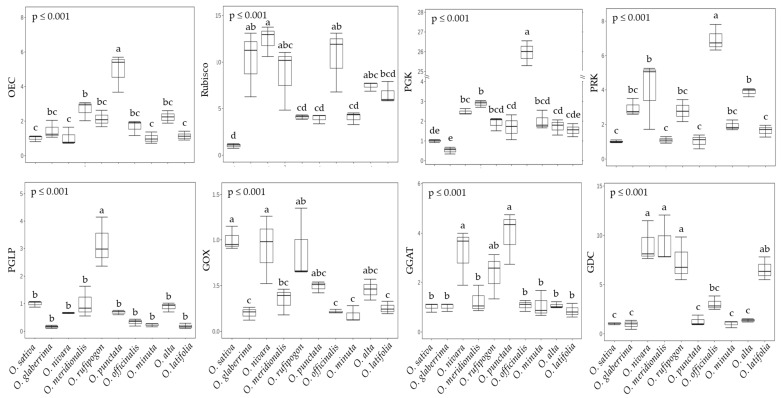
Comparison of relative gene expression encoding several major enzymes associated with photosynthesis and photorespiration between wild rice species at 60 DAT (mean of three replications ± S.D.). Means followed by different letters in the same raw are significantly different at *p* < 0.05 according to Tukey’s HSD test. OEC, oxygen-evolving complex; Rubisco, rubisco-1,5-bisphosphate carboxylase/oxygenase; PGK, phosphoglycerate kinase; PRK, phosphoribulo kinase; PGLP, phosphoglycolate phosphatase; GOX, glycolate oxidase; GGAT, glutamate glyoxylate aminotransferase; GDC, glycine decarboxylase.

**Figure 5 ijms-23-08901-f005:**
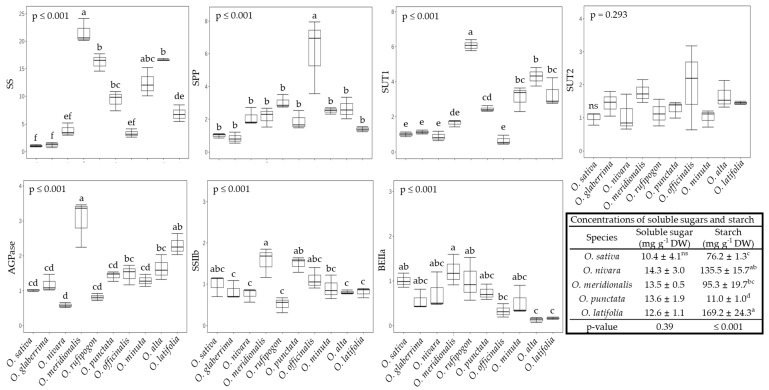
Comparison of relative gene expression encoding several major enzymes associated with sucrose and starch biosynthesis and sucrose transport and the concentrations of total soluble sugars and starch between wild rice species at 60 DAT (mean of three replications ± S.D.). Means followed by different letters in the same raw are significantly different at *p* < 0.05 according to Tukey’s HSD test. SS, sucrose synthase; SPP, sucrose phosphatase; SUT1 and 2, sucrose transporter 1 and 2; AGPase, ADP-glucose pyrophophorylase; SSIIb, soluble starch synthase; BEIIa, starch branching enzyme.

**Figure 6 ijms-23-08901-f006:**
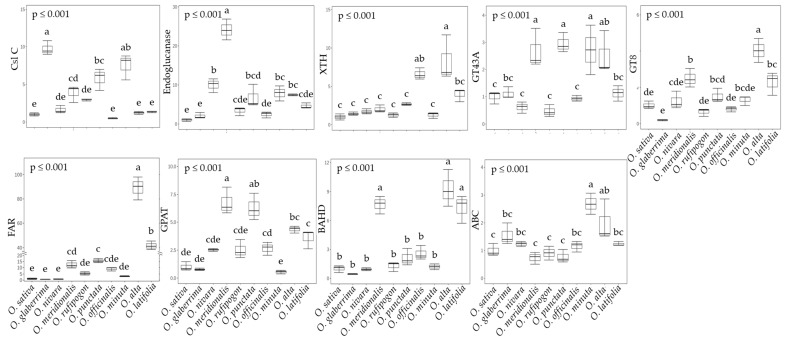
Comparison of relative gene expression encoding several major enzymes associated with primary (top) and secondary (bottom) cell wall synthesis between wild rice species at 60 DAT (mean of three replications ± S.D.). Means followed by different letters in the same raw are significantly different at *p* < 0.05 according to Tukey’s HSD test. CslC, cellulose-synthase-like C; Endoglucanase; XTH, endotransglucosylase/hydrolase; GT43A, glycosyltransferase; GT8, glycosyltransferase; FAR, fatty acyl reductase; GPAT, glycerol-3-P acyltransferase; BAHD, BAHD acyltransferase; ABC, ATP-binding cassette transporter.

**Figure 7 ijms-23-08901-f007:**
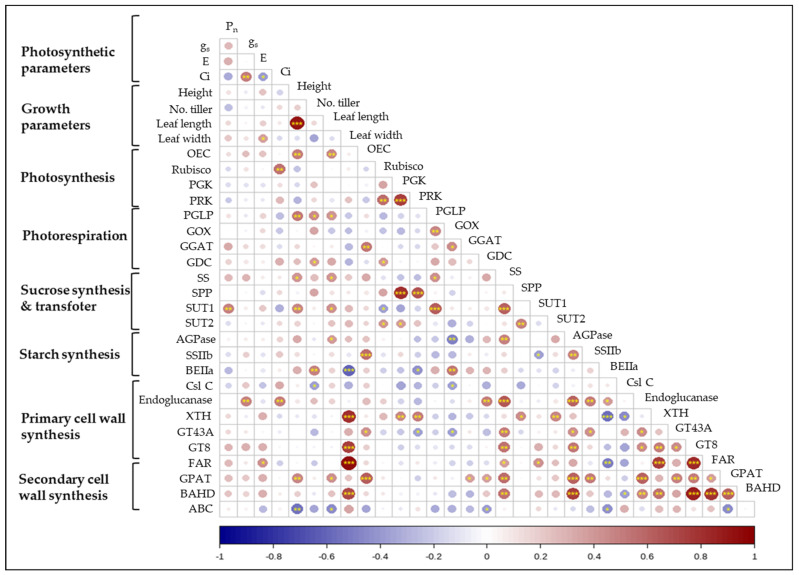
Pearson’s correlation coefficient between genes in 10 *Oryza* species datasets. Red and blue colors represent positive and negative correlations, respectively. An asterisk, *, **, and ***, indicates the significance at *p* < 0.05, 0.01, and 0.001, respectively. OEC, oxygen-evolving complex; Rubisco, rubisco-1,5-bisphosphate carboxylase/oxygenase; PGK, phosphoglycerate kinase; PRK, phosphoribulo kinase; PGLP, phosphoglycolate phosphatase; GOX, glycolate oxidase; GGAT, glutamate glyoxylate aminotransferase; GDC, glycine decarboxylase; SS, sucrose synthase; SPP, sucrose phosphatase; SUT1 and 2, sucrose transporter 1 and 2; AGPase, ADP-glucose pyrophophorylase; SSIIb, soluble starch synthase; BEIIa, starch branching enzyme; CslC, cellulose-synthase-like C; Endoglucanase; XTH, endotransglucosylase/hydrolase; GT43A, glycosyltransferase; GT8, glycosyltransferase; FAR, fatty acyl reductase; GPAT, glycerol-3-P acyltransferase; BAHD, BAHD acyltransferase; ABC, ATP-binding cassette transporter.

**Figure 8 ijms-23-08901-f008:**
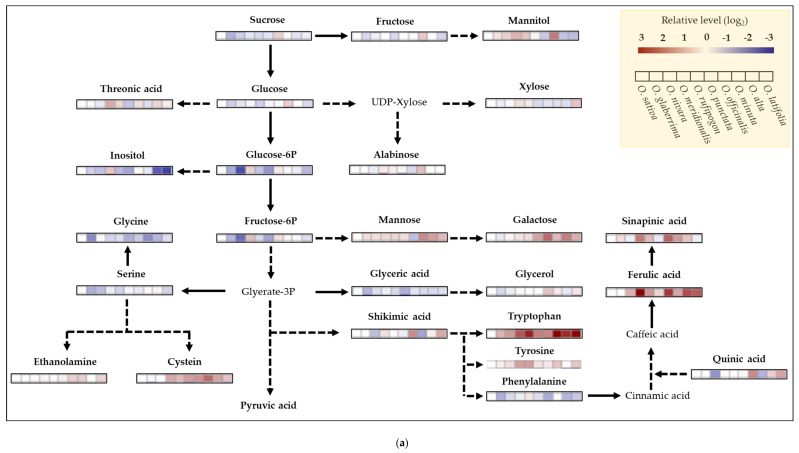
Mapping of targeted metabolites from the upper fully expanded leaves of rice species using GC-TOFMS (n = 3). Each color indicates relatively higher (red) and lower (blue) compared to *O. sativa s sp. japonica* cv. Saechucheongbyeo (RDA-Suwon433).

## Data Availability

Not applicable.

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
