# Peer review of "Transcriptional Comparison of Genes Associated with Photosynthesis, Photorespiration, and Photo-Assimilate Allocation and Metabolic Profiling of Rice Species"

_ijms, 2022, doi:10.3390/ijms23168901_

Round 1
Reviewer 1 Report
(1) Why did you use these ten rice species? There are many other sequenced rice species, e.g., Oryzaaustraliensis (Phillips AL et al. 2022), Oryza barthii (Zhang et al. 2014, Stein JC et al.2018), and Oryza brachyantha (Chen et al. 2013). Besides, why did you not use Oryza sativa ssp. indica in your study?
(2) What about the phylogenetic relationship of these ten rice species? Phylogenetic relationship will be helpful for readers. Furthermore, what about the yield and quality about these ten rice species? These are very important parameters.
(3) Net photosynthetic rate was very low in your study. In rice, Pn is usually more than 20. Is there any stresses or others in your study?
(4) Some gene expression levels were different in rice. What about the genomic similarity of there genes? Are these some key amino acids or nucleotide acids associated with gene expression difference?
(5) Your study used Pearson’s correlation coefficient. What about Spearman or Kendell correlation coefficient?
Author Response
Dear Reviewer
We are grateful to give us valuable comments and suggestions for improving our MS. We have tried to revise our MS according to your suggestion. We hope our revised MS was addressed to what you suggested.
Once again, thank you for your kind consideration.
(1) Reviewer comment: Why did you use these ten rice species? There are many other sequenced rice species, e.g., Oryza australiensis (Phillips AL et al. 2022), Oryza barthii (Zhang et al. 2014, Stein JC et al.2018), and Oryza brachyantha (Chen et al. 2013). Besides, why did you not use Oryza sativa ssp. indica in your study?
- Author response: Thank you for your comment. We totally agree with you that there are many rice species and sativa is divided into ssp, japonica and indica. Currently, 27 wild rice species were found from the huge number of research. In our study, we focused on comparing between the relatively close rice species from rice phylogenetic tree and, moreover, it was not easy to obtain all rice resources from IRRI. Secondly, we just choose one species, sativa ssp. Japonica because our approach is to compare between rice species although Japonica and Indica have different characteristics. We hope you will kindly understand.
(2) What about the phylogenetic relationship of these ten rice species? Phylogenetic relationship will be helpful for readers. Furthermore, what about the yield and quality about these ten rice species? These are very important parameters.
- Author response: Thank you for your suggestion. We will add the phylogenetic tree of rice species with a highlight of those used for this study in Supplementary data set. We agree on your consideration about yield and quality of rice species. Most of wild species showed higher shattering than we expected, and so, we didn’t want to provide incorrect data for the readers. According to your suggestion, our further research, currently under the experiment, will measure biomass, and production and quality of seed.
(3) Net photosynthetic rate was very low in your study. In rice, Pn is usually more than 20. Is there any stresses or others in your study?
- Author response: Thank you for your comment. According to the previous study (Zhao et al., 2010; Mathan et al, 2021), we also expected Pn ranges between 15 to 30, however, our experiment showed the half in spite of repeated measurements. Even, in order to avoid any incorrect measurement, we measured between 10:00 ~ 12:00 of sunny day (about 1000 μmol m-2 s-1 of PPFD). Once again, thank you for valuable comment.
(4) Some gene expression levels were different in rice. What about the genomic similarity of there genes? Are these some key amino acids or nucleotide acids associated with gene expression difference?
- Author response: Thank you for your comment. Frankly, we are very sorry not to understand what you want to be explained.
(5) Your study used Pearson’s correlation coefficient. What about Spearman or Kendell correlation coefficient?
- Author response: Thank you for your suggestion. As you know, the difference between Person and Spearman/Kendell is linear or monotonic, respectively. Under our knowledge about correlation analysis, Person provides the degree of strong or weak of positive or negative relationship, whereas Spearman and Kendell just show the correlation between both data. We want to know the degree between two data, and used Pearson’s correlation.
Reviewer 2 Report
The paper "Transcriptional Comparison of Genes Associated with Photosynthesis, Photorespiration and Photo-assimilate Allocation and Metabolic Profiling of Rice species" by Joo et al. contains valuable information to improve the production or quality of rice through breeding with wild species of Oryza. It provides a good information about photosynthetic capacity and some metabolic features important to improve production and/or resistance of rice. The main problem I find in the current version is the big number of abbreviations , which together to the number of species studied make difficult reading of the article. If possible, I reccomend authors to reduce abbreviations, as much as possible and include full-names in figure legends. Figure legends might be completely auto-explainable. It should also facilitate reading to group the species according to their main features and following an strict order when described their differential characteristics.
Author Response
Dear Reviewer
We are grateful to give us valuable comments and suggestions for improving our MS. We have tried to revise our MS according to your suggestion. We hope our revised MS was addressed to what you suggested.
Once again, thank you for your kind consideration.
The paper "Transcriptional Comparison of Genes Associated with Photosynthesis, Photorespiration and Photo-assimilate Allocation and Metabolic Profiling of Rice species" by Joo et al. contains valuable information to improve the production or quality of rice through breeding with wild species of Oryza.
It provides a good information about photosynthetic capacity and some metabolic features important to improve production and/or resistance of rice.
The main problem I find in the current version is the big number of abbreviations, which together to the number of species studied make difficult reading of the article.
If possible, I recommend authors to reduce abbreviations, as much as possible and include full-names in figure legends.
Figure legends might be completely auto-explainable. It should also facilitate reading to group the species according to their main features and following an strict order when described their differential characteristics.
- Author response: Thank you for your suggestion. We described full name of enzymes at Fig. 4, 5, 6 and 7. Once again, thank you for valuable comment.
Reviewer 3 Report
In the manuscript titled with "Transcriptional Comparison of Genes Associated with Photosynthesis, Photorespiration and Photo-assimilate Allocation and Metabolic Profiling of Rice species", the authors descripted an extensive research on the wild rice compared with cultivated rice to provide some guidelines for increasing crop yield in the future. overall, the manuscript presented a much needed information on the wild rice phenotype studies with regarding to their genome type, photosynthesis rate etc. However, the quality of the manuscript presentation still needs a lot of work to make it more accessible to the readers.
For example: The authors state that (line 72), we will compare wild and cultivated rice in this paper. However, when they first introduced those, there is no mentioning which one is wild or cultivated.
Line 93, the higher biomass-producing species... where is the data for high biomass producing? literature own data? Please refer to that.
Line 113 , the Pn number is one of the most important data the authors try to compare between wild and cultivated rice, however, it is in supplement. Please move to main text.
Line 119, WUE indicating more than 3. If the number is larger than 3, it is not indicating. It is what it is. Correct your wording or explain what this number is.
Line 126. the expression level of the BB genotype. Expression level of what protein in BB genotype? right?
Line 136-138, the authors seem quoting some quite contradicting results in literature. However, there is no references! That is not acceptable.
And there are many more similar situations. The manuscript needs more editing works. Otherwise, it is a fairly good paper.
Author Response
Dear Reviewer
We are grateful to give us valuable comments and suggestions for improving our MS. We have tried to revise our MS according to your suggestion. We hope our revised MS was addressed to what you suggested.
Once again, thank you for your kind consideration.
In the manuscript titled with "Transcriptional Comparison of Genes Associated with Photosynthesis, Photorespiration and Photo-assimilate Allocation and Metabolic Profiling of Rice species", the authors descripted an extensive research on the wild rice compared with cultivated rice to provide some guidelines for increasing crop yield in the future. overall, the manuscript presented a much needed information on the wild rice phenotype studies with regarding to their genome type, photosynthesis rate etc. However, the quality of the manuscript presentation still needs a lot of work to make it more accessible to the readers.
For example: The authors state that (line 72), we will compare wild and cultivated rice in this paper. However, when they first introduced those, there is no mentioning which one is wild or cultivated.
- Author response: Thank you for your comment. According to your comment, we changed the sentence as follow;
u rice species (wild and cultivated) " rice species including the cultivated, O. sativa ssp. Japonica cv. Saechucheongbyeo (RDA-Suwon433)
Line 93, the higher biomass-producing species... where is the data for high biomass producing? literature own data? Please refer to that.
- Author response: Thank you for your comment. With accepting your point, we concluded the expression ‘the higher biomass-producing species’ without any measurement was inappropriate, and we deleted that sentence in order to avoid providing incorrect information for the readers. Once again, thank you for valuable comment.
Line 113 , the Pn number is one of the most important data the authors try to compare between wild and cultivated rice, however, it is in supplement. Please move to main text.
- Author response: Thank you for your suggestion. In fact, we showed the Pn value in Fig. 3. Additionally, we compared the difference between the continents and between the genotypes and the data was included in Table S1-4 of S Data Set 1.
Line 119, WUE indicating more than 3. If the number is larger than 3, it is not indicating. It is what it is. Correct your wording or explain what this number is.
- Author response: Thank you for your comment. According to Haritha et al (2017), most of rice analyzed showed WUE (Pn/E), which is between 1.5 and 3.0, and Shanmugam et al (2021) reported the WUE was higher in wild species (more than 2) compared to cultivated (between 0.5~1.0). In our study, O. minuta showed more than 3 of WUE, and we described like that. However, we changed the sentence as follow and an explanation why O. minuta is higher WUE was handled in Discussion.
u The WUE was the highest in BBCC (O. minuta). O. minuta represented significant higher water use efficiency (WUE, Pn/E), indicating more than 3, compared to other rice relatives. " O. minuta (BBCC genotype) was the highest WUE (Pn/E), indicating more than 3.
Line 126. the expression level of the BB genotype. Expression level of what protein in BB genotype? right?
- Author response: Thank you for your comment. In order to avoid inconvenience of the readers, we changed the sentence as follow based on reviewer’s suggestion.
u Namely, the expression level of the BB genotype (O. punctata) was revealed to be 3 ~ 5 times greater than other groups. " Namely, the expression level of gene (Os07g0544800) encoding OEC in O. punctata (BB genotype) was revealed to be 3 ~ 5 times greater than other groups.
Line 136-138, the authors seem quoting some quite contradicting results in literature. However, there is no references! That is not acceptable.
- Author response: Thank you for your comment. We added some citations which GDC, one key enzyme involved in photorespiration, affected photosynthesis (Timm et al., 2013; Zhou et al., 2013). Once again, thank you for valuable point out.
And there are many more similar situations. The manuscript needs more editing works. Otherwise, it is a fairly good paper.